# Challenges Facing Two Outbreaks of Carbapenem-Resistant *Acinetobacter baumannii*: From Cefiderocol Susceptibility Testing to the Emergence of Cefiderocol-Resistant Mutants

**DOI:** 10.3390/antibiotics13080784

**Published:** 2024-08-21

**Authors:** Montserrat Rodríguez-Aguirregabiria, Fernando Lázaro-Perona, Juana Begoña Cacho-Calvo, Mª Soledad Arellano-Serrano, Juan Carlos Ramos-Ramos, Eduardo Rubio-Mora, Mariana Díaz-Almirón, Mª José Asensio-Martín

**Affiliations:** 1Critical Care Department, Hospital Universitario La Paz, IdiPAZ, 28046 Madrid, Spain; mariasoledad.arellano@salud.madrid.org (M.S.A.-S.); mjose.asensio@salud.madrid.org (M.J.A.-M.); 2Microbiology Department, Hospital Universitario La Paz, 28046 Madrid, Spainjuanabegona.cacho@salud.madrid.org (J.B.C.-C.); ermora@salud.madrid.org (E.R.-M.); 3Internal Medicine Department, Infectious Diseases Unit, Hospital Universitario La Paz, CIBERINFEC, IdiPAZ, 28046 Madrid, Spain; juancarlos.ramos@salud.madrid.org; 4Research Unit, Hospital La Paz Institute for Health Research, IdiPAZ, 28046 Madrid, Spain; mariana.diaz@salud.madrid.org

**Keywords:** *Acinetobacter baumannii*, carbapenem-resistant, outbreak, cefiderocol

## Abstract

Carbapenem-resistant *Acinetobacter baumannii* (CRAB) infections are associated with poor outcomes depending on patient’s conditions, clinical severity and type of infection, and treatment is challenging given the limited therapeutic options available. The aim of this study was to describe the clinical and microbiological characteristics of two outbreaks caused by CRAB in an intensive care unit (ICU). In addition, the mechanisms of resistance detected in these strains and the treatment chosen according to the available therapeutic options were analyzed. Overall, 28 patients were included. Ten patients (35.71%) had ventilator-associated pneumonia (VAP), ten (35.71%) had a bloodstream infection (BSI), and eight (28.57%) were only colonized. Recurrent infection occurred in 25% (5/20) of infected patients. Two different strains of *A. baumannii* were isolated from the index patient of the first outbreak. The first strain belonged to the ST85 and carried the *bla*_NDM-1_ carbapenemase gene, while the second belonged to the ST2 and carried *bla*_OXA-23,_ and *bla*_OXA-66_ carbapenemase genes. The phylogenetic analysis revealed that the ST2 strain was the cause of the major outbreak, and mutations in the AmpC gene were related to progressive increasing minimum inhibitory concentration (MIC) and finally, cefiderocol-resistance in one strain. The CRAB isolates from the second outbreak were also identified as ST2. Cefiderocol-resistant strains tests identified by the disc diffusion method were involved in 24% (6/25) of nosocomial infections. Using broth microdilution (BMD) ComASP^®^ only, 33.3% (2/6) of these strains were cefiderocol-resistant. All-cause ICU mortality was 21.4%. Conclusions: Cefiderocol is the first approved siderophore cephalosporin for the treatment of CRAB infections. Cefiderocol-resistant strains were related with *bla*_NDM-1_ carbapenemase and mutations in the AmpC gene. Cefiderocol-resistant strains or that cannot be properly interpreted by disk diffusion, should be retested using BMD for definitive categorization.

## 1. Introduction

*Acinetobacter baumannii* complex is an opportunistic pathogen that has successfully developed multiple antibiotic resistance mechanisms and frequently causes outbreaks in healthcare institutions and hospitals [1]. *A. baumannii* poses a significant threat to immunocompromised or chronically ill patients, particularly in intensive care units (ICUs) due to high exposure to invasive procedures and prolonged antibiotic therapy [2,3]. *A. baumannii* can easily survive on both moist and dry surfaces. In addition, *A. baumannii* can also colonize the skin, respiratory tract, oral cavity, and other sections of the human body [4]. Differentiating between colonization and acute infection in critically ill patients is challenging [5].

Resistance mechanisms in *A. baumannii* include reduced membrane permeability, antibiotic efflux, genetic mutation, post-translational modifications, and antibiotic inactivation via beta-lactamases [6]. Carbapenem resistance is mainly due to OXA-type Class-D enzymes (OXA-27, OXA-49, OXA-25, OXA-26, and OXA-40) and less frequently to metallo-β-lactamases (MBLs) such as VIM, IMP, and NDM-1. Additionally, chromosomal cephalosporinase (e.g., AmpC, plasmid-mediated β-lactamases such as TEM-1, TEM-2, OXA-21, and OXA-37), and extended-spectrum b-lactamases (ESBLs) are frequently expressed [7].

The most common clinical manifestations of *A. baumannii* are nosocomial pneumonia and bloodstream infection (BSI). Although *A. baumannii* has been considered a pathogen with limited virulence, invasive infections, especially those caused by multi-drug resistant strains, are associated with increased morbidity and mortality in predisposed patients [8]. In the EUROBACT-II study, 11.9% of hospital-acquired BSI were caused by *A. baumannii* and 84.6% were resistant to carbapenems [9]. Falcone et al. recently reported a 43.2% thirty-day mortality in patients with CRAB bacteremia, and the attributable mortality rate was 16% [10]. *A. baumannii* is usually a cause of ventilator-associated pneumonia (VAP) with mortality ranging from 30 to 70% [11,12]. CRAB infections have become especially difficult to treat owing to the paucity of therapeutic options. There are no conclusive data regarding the optimal antibiotic therapy against CRAB infections [12,13].

The aim of this study was to describe the clinical and microbiological characteristics of two outbreaks caused by CRAB in an ICU. In addition, the mechanisms of resistance detected in these strains and the treatment chosen according to the available therapeutic options were analyzed.

## 2. Materials and Methods

### 2.1. Study Design and Patient Selection

This was a retrospective, observational study conducted in a major teaching hospital. Patients colonized or infected by CRAB who were admitted consecutively in the ICU from October 2022 to November 2023 were included. 

The inclusion criteria were: (1) age ≥ 18 years; (2) positive surveillance cultures for CRAB; (3) blood culture or respiratory tract culture positive for CRAB. Polymicrobial etiology was not excluded. All episodes of CRAB infections were reported for each patient. All patients were managed by the same team of physicians. Antimicrobial therapies were selected according to the clinical judgement of the team of physicians and infectious disease specialists. All data were extracted retrospectively. 

Medical records were reviewed to extract clinical information. The following data were recorded: demographics, reason for admission, Charlson comorbidity index, and comorbid conditions. Immunodepression included advanced solid cancer, solid organ transplantation, hemato-oncological patients, corticosteroid therapy or other chronic immunosuppressive therapies, and HIV infection. In addition, the following data were registered: Acute Physiology and Chronic Health Evaluation (APACHE) II score, CRAB colonization or infection, source of infection, the empirical and targeted antimicrobial therapy, severity with the Sequential Organ Failure Assessment score (SOFA) at the time of infection, development of septic shock, life-supporting therapies such as mechanical ventilation (MV), continuous renal replacement therapy (CRRT), and extracorporeal membrane oxygenation (ECMO) carried out during hospitalization, and length of stay in the intensive care unit (ICU) and hospital. The Abbreviated Burn Severity Index (ABSI) and Total Burned Body Surface area (TBSA), and early surgical intervention by escharotomy were recorded in burn patients. The development of resistance to antibiotic treatment was carefully monitored. The decision to withhold or withdraw life-sustaining treatment because of underlying conditions or the futility of invasive treatment was reviewed. The occurrence of adverse events during antibiotic treatment, especially nephrotoxicity, was noted. Acute kidney injury (AKI) was defined according to the Second International Consensus Conference of the Acute Dialysis Quality Initiative (ADQI) group [14]. Early mortality, within 7 days after the diagnosis and related to the infection, and all-cause mortality in ICU were assessed.

### 2.2. Definitions

According to CDC/NHSC criteria, infections were classified into the following categories: ventilator-associated pneumonia and bloodstream infections [15]. The onset of bacteriemia was defined as the day when the blood culture that eventually yielded *A. baumannii* was obtained. Episodes of bloodstream infections were considered acquired in the ICU if they appeared within 48 h after ICU admission. The severity of infections was assessed according to the Third International Consensus Definitions for Sepsis and Septic Shock [16]. The length of hospital and ICU stay was calculated as the number of days from the date of admission to the date of discharge or death. The Charlson Comorbidity Index was calculated based on previous definitions [17]. CRAB colonization was evaluated weekly through active surveillance in ICU patients [18]. Clinical cure was defined as the resolution of signs and symptoms of infection. Relapse or recurrence was defined as a subsequent CRAB infection requiring antimicrobial treatment. Progression, recurrence, or relapse of nosocomial pneumonia were considered treatment failure. 

### 2.3. Bacterial Isolates Identification and Susceptibility Testing

#### 2.3.1. Strain

All clinical isolates from patients were identified using MALDI TOF (Bruker Daltonics GmbH & Co. KG, Bremen, Germany). The MicroScan WalkAway plus System (Beckman Coulter, Brea, CA, USA) was used for routine antimicrobial susceptibility tests (AST). The results of AST were interpreted according to the European Committee on Antimicrobial Susceptibility Testing (EUCAST) breakpoints. 

#### 2.3.2. Cefiderocol Susceptibility Test

The first isolate of CRAB from each patient was tested for cefiderocol susceptibility. The cefiderocol susceptibility tests were performed by the disc diffusion method (Liofilchem, Roseto degli Abruzzi, Italy) on Mueller–Hinton medium plates (MH agar plates), and new broth microdilution (BMD) commercially available tests, ComASP^®^ (Liofilchem, Roseto degli Abruzzi, Italy), in vitro according to the FDA drug approved label (https://www.accessdata.fda.gov/cdrh_docs/reviews/K230479.pdf, accessed on 14 July 2024). The results were interpreted according to the breakpoints proposed by EUCAST: zone diameters of ≥17 mm for the cefiderocol 30 µg disk correspond to minimum inhibitory concentration (MIC) values below the PK-PD breakpoint of S ≤ 2 mg/L (https://www.eucast.org/clinical_breakpoints, accessed on 14 July 2024). BMD ComASP^®^ was not available in our institution during the first outbreak and the strains were tested afterwards.

Synergy analyses were performed using the fixed proportion with the E-test method. The fractional inhibitory concentration (FIC) index (FICI) was calculated for the E-test by using the following formula: FICI = FIC of agent A + FIC B, where FIC A is the MIC of the combination/MIC of drug A alone, and FIC B is the MIC of the combination/MIC of drug B alone. FICI results were interpreted with the following criteria: synergy, FICI ≤ 0.5; independent interaction, FICI 0.5- ≤ 4; and antagonism, FICI > 4 [19]. 

### 2.4. Molecular Analysis

Most *A. baumannii* recovered from patients or the environment were subjected to whole genome sequencing during the outbreak. DNA was extracted with the DNeasy^®^ blood&Tissue kit (Qiagen GmbH, Hilden, Germany) and sequenced in the Ion GeneStudio S5 system (Thermo Fisher Scientific, Waltham, MA, USA) using the NEBNext^®^ Fast DNA Library Prep Set for Ion Torrent™ (New England Biolabs, Ipswich, MA, USA). Genome assembly was performed using spades v3.14.1 (https://github.com/ablab/spades, accessed on 14 July 2024) and single nucleotide polymorphisms (SNP) analysis with Snippy v4.6.0 (https://github.com/tseemann/snippy, accessed on 14 July 2024). Phylogenetic analysis was carried on using FastTree 2.0.0 (http://www.microbesonline.org/fasttree, accessed on 14 July 2024). Analysis of resistance genes and plasmids was performed using ABRicate v1.0.1 (https://github.com/tseemann/abricate, accessed on 21.8.24). MLST was determined with the mlst v.2.23.0 software (https://github.com/tseemann/mlst, accessed on 14 July 2024). Phylogenetic results were continuously reported to the Preventive Medicine department.

### 2.5. Statistical Analysis

A descriptive univariate analysis was carried out for all study variables. Frequency results were expressed as absolute and relative frequencies. Continuous variables were expressed with the main measures of dispersion (mean, standard deviation, median, and interquartile range). Comparative analysis was performed with the Pearson Chi-Square test (or Fisher’s exact test for 2 × 2 tables or likelihood ratio in mXn tables, if necessary) or with non-parametric equivalents Mann–Whitney U test in the case of quantitative variables. The type I error rate (alpha) was set to 5% (two-sided).

## 3. Results

### 3.1. Study Population

The index case was a burn patient coming from Algeria. One hundred and seventy-one patients were admitted to the ICU during the first outbreak (from October 2022 to June 2023). Eighteen (10.52%) patients were infected by CRAB and seven (4%) became colonized. The second outbreak was triggered in October 2023, with two patients also burned and coming from another institution. No new patients were infected by CRAB and only one patient became colonized (Figure 1).

Both outbreaks affected 28 critically ill patients with APACHE II: 14.1 (SD 5.4) points, who had long hospital stays, underwent mostly invasive procedures, and half required hemodynamic support. The reason for admission was burn patients (60.71%) and medical disease (21.42%). Immunosuppressed patients accounted for 21.42%. The most relevant characteristics of patients are described below (Table 1).

Overall, 20 patients (71.42%) were infected, and 8 were colonized by CRAB (28.57%). Most of the infected patients had severe burns (70%) with greater and depth skin damage. Abbreviated Burn Severity Index (ABSI) and Total Body Surface Area (TBSA) affected were higher in burned patients with a nosocomial CRAB infection. All burn patients requiring urgent escharotomy became infected. 

All patients with continuous renal replacement therapy (CRRT) developed an invasive infection. In one patient, ECMO was previously implanted for acute respiratory distress syndrome due to *Staphylococcus aureus* nosocomial pneumonia, and the other case was because of a VAP caused by CRAB. The differences observed between colonized and infected patients are described below (Table 2).

All-cause ICU mortality was 21.4% (6/28 patients). In deceased patients, APACHE II was higher, and invasive procedures were more frequent. Mechanical ventilation was required in 80.3% of these patients, and 50% had to undergo CRRT. Fifty percent (3/6) were burn patients with a higher ABSI and TBSA affected. No colonized patient died during the study period (Table 3).

### 3.2. Characteristics, Treatments, and Outcomes of Patients with CRAB Infections

Overall, 20 patients were infected. Ten (50%) developed a VAP, and the other 50% had a BSI. Catheter infections were the cause of 40% (4/10) of bacteremia, and 20% (2/10) were bacteremia of unknown origin. In 40% (4/10) of episodes, the source was skin and soft tissue infections or related to surgical procedures in burn patients. Fifty percent of the first CRAB infections were polymicrobial, and *Pseudomonas aeruginosa* (4/10) and *S. aureus* (2/10) were the most frequently isolated microorganisms (details in Appendix A). Three patients developed candidemia and one had a probable invasive pulmonary aspergillosis.

All deceased patients developed septic shock during CRAB infection, five patients (83.33%) due to a VAP, and one (16.66%) had a bacteremia. The SOFA score was higher, and in 50% (3/6) of cases, the empirical treatment of CRAB infections was inappropriate. 

As for targeted therapy, cefiderocol was administered as monotherapy 40% (8/20) and in combination 25% (5/20) with other antimicrobials, mainly colistin and sulbactam. Eighty percent (8/10) of tested strains were resistant to tigecycline following CLSI criteria. Treatment was tailored according to the microbiological report. The clinical success rate in BSI was 100% (10/10). Clinical success in VAP was 77.7% (7/9); one patient did not receive antibiotic treatment because it was decided to discontinue life support treatment due to the severity of the burn injury (details in Appendix A).

Three patients with septic shock in the course of CRAB infections who received colistin as part of antibiotic treatment developed acute kidney injury (AKI). Five patients had a recurrent CRAB infection; three were treated with cefiderocol, and two with colistin in monotherapy. In only two cases, death was considered related to infection (both patients with recurrent VAP). Targeted treatment is described below (Table 4).

### 3.3. Cefiderocol Susceptibility Test

Susceptibility testing for cefiderocol was performed in the first isolate from 20 infected patients by disc diffusion method. Of these, four (20%) were resistant to cefiderocol and ComASP^®^ was performed: three strains were susceptible with MIC = 2 mg/L, and one was resistant with MIC = 16 mg/L. On this last strain, synergistic activity with ampicillin/sulbactam was found with a FICI = 0.019. 

Five patients (25%) had a clinical and microbiological recurrence. The tests of susceptibility for cefiderocol were performed in three patients; two isolates were resistant to disc diffusion, and only one had a MIC = 4 mg/L (details in Appendix A).

### 3.4. Phylogenetic Analysis of the Outbreak

Twenty-four clinical isolates (one per patient) and eight environmental isolates were subjected to whole genome sequencing. During the first outbreak, the initial analysis discovered that the index patient was initially colonized by two different strains of *A. baumannii*. The first strain belonged to the ST85 and carried the *bla*_NDM-1_ carbapenemase gene while the second belonged to the ST2 and carried *bla*_OXA-23_, as well as *bla*_OXA-66_ carbapenemase genes.

The phylogenetic analysis revealed that the ST85 NDM-1-producer strain was transmitted to a second patient who was only colonized, while the ST2 strain was the cause of the major outbreak. This strain was identified in all but one of the studied patients from the first outbreak and multiple inanimate objects. A single nucleotide polymorphisms (SNP) analysis of cefiderocol-resistant strains showed that all of them presented mutations in the AmpC gene, which was related to progressively increasing MIC, and to cefiderocol resistance in one strain. That mutation was not present in the cefiderocol-susceptible strain isolated from the index patient.

The three CRAB isolates from the second outbreak were also identified as ST2, but in this case, phylogenetic analysis showed that the three isolates were not part of the previous outbreak and were not related to each other.

## 4. Discussion

*Acinetobacter baumannii* has emerged as a major nosocomial pathogen exhibiting high rates of resistance to clinically relevant antibiotics. It frequently causes outbreaks in healthcare institutions and hospitals. The index case was a critically burned patient with multifocal colonization and a CRAB infection. Main infection control measures implemented throughout the first outbreak included hand hygiene, contact isolation precautions, environmental cleaning, an active surveillance program to identify asymptomatic carriage, and finally patient cohorting in a contained area and using dedicated staff [20]. Lessons learned from this complex scenario helped to contain the second outbreak in which only one patient ended up colonized.

In our study, 20 patients developed a CRAB infection, and 70% were patients with severe burn injuries. Burn patients are at particularly high risk for invasive infection due to the impairment of host immunity and loss of skin barrier function, and infection attributable mortality ranges from 50 to 75% [21]. In our research, half of nosocomial infections were polymicrobial, with *P. aeruginosa* and *S. aureus* being the most frequently isolated microorganisms. Therefore, it is difficult to attribute a prognostic impact to the isolation of CRAB. When CRAB is isolated from a non-sterile site, it may represent colonization or infection, and distinguishing between the two is often a challenge [22]. Data from large retrospective studies identifies BSI as a negative predictor of clinical outcomes in burn patients [23]. Risk factors associated with mortality include age, SOFA score at the onset of bacteremia, and TBSA [24]. This impact could not be assessed in our study given the small sample size. Only one patient with CRAB bacteremia died. All-cause mortality in the ICU was 21.4%, but there were no infection-related deaths within seven days of diagnosis. ICU mortality was due to the severity of burn injuries, or the decision to withhold or withdraw life-sustaining treatment in patients admitted with life-threatening conditions and the futility of invasive measures, and palliative care was offered [25,26].

Cefiderocol-resistant was found in one ST85 NDM-1 producer strain and was transmitted to a second patient who was only colonized. ST2 strains carrying *bla*_OXA-23_, as well as *bla*_OXA-66_ carbapenemase genes, were the cause of the major outbreak; in addition, mutations in the AmpC gene were detected. Multidrug-resistant strains of this sequence type have been the cause of numerous ICU outbreaks worldwide [27,28,29]. The temporal and genomic relationship between the cases suggests that cefiderocol-resistant CRAB (MIC increased to 4 mg/L in one isolated) was selected, probably by increasing the use of cefiderocol in the ICU. In this context, it is possible that a subtherapeutic concentration of the antibiotic in the skin and soft tissue facilitated the progressive emergence of resistance. Some inoculum effects cannot be ruled out, especially in cases of a high bacterial inoculum infection such as VAP, leading to the selection of resistant bacteria [30]. In the second outbreak, two different CRAB ST2 strains were found, though in this case, the transmission was contained. Furthermore, the genomic analysis allowed for ruling out a possible reemergence of the previous outbreak. 

Resistance to cefiderocol has been reported. In a systematic review, cefiderocol non-susceptibility was especially higher among NDM-producing *A. baumannii* isolates [31]. With regard to AmpC, in vitro studies with mutational analysis have found important amino acid changes in strains not susceptible to cefiderocol [32]. For *Enterobacterales*-producing class C β-lactamases, the selection of co-resistance to cefiderocol and other antimicrobials following cefepime treatment highlights the potential impact of mutational resistance in AmpC [33]. A recent study found treatment-emergent resistance and interpatient transmission of cefiderocol-resistant *A. baumannii* harbored disrupted *pir*A and *piu*A genes that were not disrupted among susceptible isolates. In this report of 11 critically ill burn patients, all isolates were identified as *A. baumannii* sequence type ST2, one of the major clones widely distributed in the United States. Each isolate harbored *bla*_oxa-23_, *bla*_oxa-66_, and *bla*_ADC-73_ [34]. OXA-_23_-producing *A. baumannii* isolates have been reported in several countries, suggesting a wide distribution as well as from Spain [35,36].

In our study, treatment was tailored according to susceptibility testing for cefiderocol by disc diffusion method, and six isolates were resistant. However, using BMD ComASP^®^ test, resistance was only confirmed in two strains with MIC ≥ 4 mg/L, and four strains were susceptible with high MIC = 2 mg/L. This circumstance may raise the need for a higher dose of antibiotics in certain common clinical scenarios in critically ill patients. 

For *Acinetobacter* spp., there is as yet insufficient evidence to determine clinical breakpoints for cefiderocol. For these, EUCAST has determined a zone diameter that will exclude isolates with MIC values clearly above the PK-PD breakpoint (https://www.eucast.org/fileadmin/src/media/PDFs/EUCAST_files/Disk_criteria/Validation_2024/Acinetobacter_v_5.0_January_2024.pdf, accessed on 14 July 2024). Furthermore, the accuracy and reproducibility of cefiderocol testing results by disk diffusion and broth microdilution are markedly impacted by iron concentration and inoculum preparation, and may vary by disk and media manufacturer. Depending on the type of variance observed, false resistant or false susceptible results may occur (https://clsi.org/media/a5ndfge2/ast_newsletter23_final-24.pdf, accessed on 14 July 2024)

The reference method for cefiderocol antimicrobial susceptibility testing is BMD with iron-depleted Mueller–Hinton medium (ID-MH), whereas breakpoints recommended for disk diffusion are based on MH-agar plates. Recent research comparing the performance of different methods for testing in vitro activity of cefiderocol in 100 CRAB isolates found that disk diffusion and E-test on ID-MH agar plates exhibit higher diagnostic performance than on MH-agar plates and the commercial BMD methods. ComASP^®^ showed 76% essential agreement with the standard BMD method [37]. On a limited number of isolates, the ComASP^®^ microdilution panel was a valid method to determine cefiderocol MIC on isolates for which the disk diffusion results were uninterpretable, and the combination of both tests was an optimal approach to overcome the challenge of cefiderocol susceptibility testing in routine microbiology laboratories [38]. In terms of suggesting a strategy, disc diffusion could be useful for screening, and for resistant or uninterpretable isolates, a BMD test should be performed for definitive categorization [38,39,40].

The lack of harmonization between different committees with respect to cefiderocol breakpoints can lead to discrepant results [41]. The EUCAST breakpoints are generally more restrictive than those by the CLSI and allow us to detect the presence of strains with low levels of resistance. This aspect is particularly relevant when dealing with infections caused by microorganisms with very few therapeutic options, and even more so when therapeutic alternatives are considered suboptimal or associated with significant side effects, such as colistin and nephrotoxicity [42]. 

Cefiderocol is the first approved siderophore cephalosporin for the treatment of CRAB infections. The latest ESCMID guidelines recommend against cefiderocol for the treatment of infections caused by CRAB (conditional strength of recommendation, low level of evidence) [43]. The IDSA guidance document recommended that cefiderocol should be limited to the treatment of CRAB infections refractory to other antibiotics or when intolerance to other agents precludes their use [44]. Both documents agree to the treatment of severe CRAB infections using the combination of two in vitro active agents [43,44]. Data coming from the CREDIBLE-CR phase 3 study with survival rates higher with the best available therapy at the end of the study compared with cefiderocol were limited to making a different recommendation. However, it should be noted that patients assigned to the cefiderocol arm were more likely to be in the ICU at the time of randomization and have ongoing septic shock than those in the best available therapy arm, which may explain these results [45]. Data coming from real-world experience have yielded mixed results regarding the clinical effectiveness of cefiderocol for CRAB infections, especially in critically ill patients [46,47,48]. An interesting approach emerges from this study done by Dalfino et al. evaluating the effectiveness of first-line therapy with old and novel CRAB active antibiotics in VAP. Clinical failure was lower in the cefiderocol group. Timely targeted antibiotic treatment and cefiderocol-based first-line regimens strongly reduced failure risk [49]. This fact is especially relevant when treating severe infections in critically ill patients. Initial failure to choose the right antibiotic treatment often results in a worse outcome. Furthermore, a recent meta-analysis in observational studies providing proper adjustment for confounders showed that cefiderocol-based regimens were associated with a significantly lower risk of mortality in patients with CRAB infections [50].

In our study, for targeted therapy, cefiderocol was administered as monotherapy or in combination, mainly colistin and sulbactam in 64% of CRAB infections, and colistin monotherapy or in combination with 32%. The efficacy of combination treatment compared to cefiderocol monotherapy remains unresolved. Onorato et al. evaluated cefiderocol monotherapy and combination therapy among seven studies in a meta-analysis. They found a significantly lower mortality rate among patients receiving cefiderocol in monotherapy as compared to those treated with combination regimens. However, these findings were not confirmed in the sub-analysis including only patients with bloodstream infections, nor in the analysis including patients with pneumonia [51]. A recent review also found no significant difference in terms of mortality, microbiological eradication, and clinical cure between monotherapy and combination therapy [52].

In addition, the emergence of in vivo resistance to cefiderocol in the setting of CRAB infections has been reported [53]. In the Falcone et al. study, microbiological failure occurred in 17.4% of patients receiving cefiderocol versus 6.8% of those receiving colistin. The eight patients of the cefiderocol group with microbiological failure all had a BSI, and six received cefiderocol monotherapy. All relapsing *A. baumannii* strains isolated from patients receiving cefiderocol were re-tested, and four strains demonstrated resistance (MIC values range from 4 to ≥32 mg/L) [46]. In contrast, the Dalfino et al. study found lower microbiological failure in the cefiderocol group [49]. In post-hoc analysis of CREDIBLE-CR of MICs, in the cefiderocol group, 12 isolates (from 12 patients [15%]) had at least a four-fold increase in cefiderocol MIC from baseline (i.e., five for *A. baumannii*, one for *Stenotrophomonas maltophilia*, three for *Klebsiella pneumoniae*, and three for *P. aeruginosa*). Only four isolates had an MIC that increased to more than 2 µg/mL, of which three isolates had an MIC of more than 4 µg/mL [45]. In our study, we observed a temporal and genomic relationship between the strains with a progressively increasing MIC to 4 mg/L, but only in one isolate. 

An interesting approach to reduce the risk of in vivo emergence of cefiderocol-resistant strains or restore susceptibility in cefiderocol-resistant *A. baumannii* strains could be the combination with β-lactamases-inhibitor, although this hypothesis has not been confirmed by clinical trials. High-dose ampicillin-sulbactam, as a component of combination therapy, is suggested as an alternative agent for CRAB infections. Sulbactam is a competitive, irreversible β-lactamase inhibitor that, in high doses, saturates PBP1a/1b and PBP3 of *A. baumannii* isolates. Although a high proportion of isolates are currently resistant to ampicillin-sulbactam, administration is recommended even in these circumstances [5,12,44]. Human-simulated exposure of cefiderocol in combination with ceftazidime/avibactam or sulbactam (with ampicillin) resulted in potent in vivo activity against 15 carbapenem-non-susceptible *A. baumannii* including cefiderocol-non-susceptible isolates [54]. In our research, we found a synergistic effect using ampicillin-sulbactam in one patient with a cefiderocol-resistant CRAB bacteremia. The potential role of β-lactamases-inhibitors such as avibactam or sulbactam has been evaluated. Avibactam showed synergistic activity and restored in vitro cefiderocol susceptibility in some *A. baumannii* strains with non-producing MBL-carbapenemases [37]. Avibactam and durlobactam could inhibit cefiderocol hydrolysis by PER-1 [55]. However, although data about the role of β-lactamases-inhibitors in restoring available treatment for CRAB are encouraging, antagonism has also been described depending on underlying resistance mechanisms or species involved [56,57]. 

The main limitation of this study is that it is a single-center observational and retrospective study with a small sample size. Despite this, we have been able to assess the most relevant aspects of the strategy to be followed in terms of prevention, diagnosis, and treatment in a very complex clinical scenario such as infections caused by CRAB. 

We would also like to highlight that this is one of the few studies in which a phylogenetic analysis of a CRAB outbreak in critically ill patients has been performed. This has allowed us to monitor the behavior of CRAB strains over time, and to detect the underlying mechanisms related to resistance to a new antibiotic. In addition, we advise on the use of different cefiderocol susceptibility tests in order to avoid problems of accuracy in definitive categorization.

## 5. Conclusions

Infections caused by CRAB represent a major challenge in daily clinical practice due to the limited therapeutic options available and the potential impact on the prognosis of these patients. In our experience, cefiderocol could be considered a therapeutic option against CRAB infections with a better safety profile compared to other older antibiotics. Susceptibility testing to cefiderocol can be carried out by disk diffusion for screening, and for resistant or uninterpretable isolates, we suggest performing a BMD test for definitive categorization. The management of infections caused by CRAB should be approached from a global institutional perspective and requires the participation of a multidisciplinary team to design a prevention, diagnosis, and treatment strategy. 

## Figures and Tables

**Figure 1 antibiotics-13-00784-f001:**
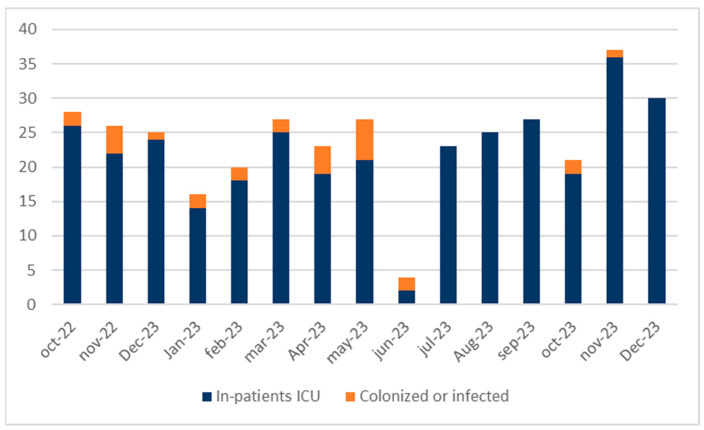
Epidemic curve of two CRAB outbreaks. Patients admitted to the ICU and not affected during the outbreaks are shown in blue. Orange represents patients colonized or infected by CRAB. First outbreak from October 2022 to June 2023. Second outbreak from October to November 2023.

**Table 1 antibiotics-13-00784-t001:** Demographics, reason for admission, baseline characteristics, life-supporting therapies, and prognosis of patients colonized or infected by CRAB in ICU during both outbreaks (from October 2022 to November 2023).

Variable	All Patients n = 28
Age in years, median (IQR)	45.5 (30.7–64.25)
Male sex	18 (64.28%)
Reason for admission	
Medical	6 (21.42%)
Trauma	1 (3.57%)
Burn	17 (60.71%)
TEN	4 (14.28%)
APACHE II, mean (SD)	14.1 (5.4)
Age-adjusted Charlson index, median (IQR)	1 (1–4)
Underlying conditions	
Diabetes mellitus	3 (10.71%)
Chronic pulmonary disease	2 (7.14%)
Chronic renal insufficiency	0
Liver cirrhosis	1 (3.57%)
HIV	1 (3.57%)
Solid cancer	3 (10.71%)
Hematologic cancer	0
Immunosuppressive drugs	3 (10.71%)
Immunocompromised (all causes)	6 (21.42%)
Length of hospital stay, median (IQR)	56.5 (37.25–92.70)
Length of ICU stay, median (IQR)	37 (19.50–66.75)
Invasive procedures	
Mechanical ventilation	18 (64.28%)
CRRT	4 (14.28%)
ECMO	2 (7.14%)
Vasopressor drugs	15 (53.6%)
Tracheostomy	13 (46.4%)
Colonized	8 (28.57%)
Infected	20 (71.42%)
First invasive infection	
VAP	10 (35.7%)
BSI	10 (35.7%)
ICU mortality	6 (21.42%)
Hospital mortality	7 (25%)
Early and related infection mortality (≤7 days)	0

TEN: Toxic epidermal necrolysis; HIV: human immunodeficiency virus; CRRT: continuous renal replacement therapy; ECMO: extracorporeal membrane oxygenation; VAP: ventilator-associated pneumonia. BSI: bloodstream infection.

**Table 2 antibiotics-13-00784-t002:** Univariate analysis between colonized and infected patients by CRAB in ICU during both outbreaks (from October 2022 to November 2023).

	Colonized (n = 8)	Infected (n = 20)	*p*
Age in years, median (IQR)	29 (22–29)	33.5 (28.2–55.2)	0.258
Male sex	6 (75%)	12 (60%)	0.669
Reason for admission			*
Medical Trauma Burn - ABSI, mean (SD) - TBSA %, mean (SD) - Escharotomy TEN	3 (37.5%) 0 3 (37.5%) 4.5 (3.5) 22.2 (15.3) 0 2 (25%)	3 (15%) 1 (5%) 14 (70%) 7.64 (2.09) 36 (15.4) 5 (25%) 2 (10%)	
APACHE II, mean (SD)	11.75 (5.6)	15.1 (5.14)	0.165
Charlson index, median (IQR)	3 (0.5–4)	0 (0–2.5)	0.150
Underlying conditions			*
Diabetes mellitus Chronic pulmonary disease Chronic renal insufficiency Liver cirrhosis HIV Solid cancer Hematological cancer Immunosuppressive drugs Immunocompromised (all causes)	2 (25%) 1 (12.7%) 0 1 (12.7%) 1 (12.5%) 1 (12.5%) 0 1 (12.7%) 3 (37.5%)	1 (5%) 1 (5%) 0 0 0 2 (10%) 0 2 (10%) 3 (15%)	
Length of ICU stay	22 (12.25–39.5)	37 (23–79.5)	0.199
Length of hospital stay	54 (43.5–150.75)	57 (31.75–98.25)	0.746
Invasive procedures			*
Mechanical ventilation CRRT ECMO Vasopressor drugs Tracheostomy	3 (37.5%) 0 0 2 (25%) 2 (25%)	15 (75%) 4 (20%) 2 (10%) 13 (65%) 11 (55%)	
ICU mortality	0	6 (30%)	*
Hospital mortality	0	7 (35%)	*

* Differences have not been assessed due to the small sample size. ABSI: Abbreviated Burn Severity Index. TBSA: Total Body Surface Area TEN: Toxic epidermal necrolysis; HIV: human immunodeficiency virus; CRRT: continuous renal replacement therapy; ECMO: extracorporeal membrane oxygenation.

**Table 3 antibiotics-13-00784-t003:** Univariate analysis of all-cause ICU mortality in colonized and infected patients by CRAB during both outbreaks (from October 2022 to November 2023).

	ICU Mortality (n = 6)	Survivors (n = 22)	*p*
Age in years (IQR)	56.3 (37–75)	45.5 (19–83)	0.259
Male sex	3 (50%)	15 (68.2%)	0.634
Reason for admission			
Medical	2 (33.3%)	4 (18.2%)	
Trauma	0	1 (4.5%)	
Burn	3 (50%)	14 (63.6%)	
- ABSI, mean (SD)	10 (1)	6.62 (2.18)	
- TBSA %, mean (SD)	42.2 (19.3)	30.47 (15.1)	0.025
- Escharotomy	2 (33%)	3 (13.6%)	0.24
TEN	1 (16.7%)	3 (13.6%)	
APACHE II, mean (SD)	19.33 (4.55)	12.73 (4.77)	0.01
Charlson index, median (IQR)	1 (0–4.5)	0.5 (0–4)	0.72
Underlying conditions			*
DM	1 (16.7%)	2 (9.1%)	
Chronic pulmonary disease	0	2 (9.1%)	
Chronic renal insufficiency	0	0	
Liver cirrhosis	0	1 (4.5%)	
HIV	0	1 (4.5%)	
Solid cancer	1 (16.7%)	2 (9.1%)	
Hematological cancer	0	0	
Immunosuppressive drugs	1 (16.7%)	2 (9.1%)	
Immunocompromised (all causes)	1 (16.7%)	5 (22.7%)	
Length of ICU stay, median (IQR)	44.5 (16.25–91.5)	37 (20.25–53.25)	0.764
Length of hospital stay, median (IQR)	44.5 (16.25–91.5)	56 (41.75–103)	0.236
Colonized	0	8 (36.3%)	0.141
Infected	6 (100%)	14 (63.6%)	
Invasive procedures			*
Mechanical ventilation	5 (83.3%)	13 (59.1%)	
CRRT	3 (50%)	1 (4.5%)	
ECMO	0	2 (9.1%)	
Vasopressor drugs	5 (83.3%)	10 (45.5%)	
Tracheostomy	3 (50%)	10 (45.5%)	
AKI	2 (33.3%)	2 (9.1%)	*

* Differences have not been assessed due to the small sample size. ABSI: Abbreviated Burn Severity Index. TBSA: Total Body Surface Area TEN: Toxic epidermal necrolysis; HIV: human immunodeficiency virus; CRRT: continuous renal replacement therapy; ECMO: extracorporeal membrane oxygenation; AKI: acute kidney injury.

**Table 4 antibiotics-13-00784-t004:** Characteristics and outcomes of patients with CRAB infections.

CRAB Infections (n = 20)	ICU Mortality (n = 6)	Survivors (n = 14)
Septic shock	6 (100%)	6 (30%)
SOFA score, median (IQR)	11 (7.5–14.25)	4.5 (3–7.25)
VAP (n = 10)	5 (83.33%)	5 (35.71%)
BSI (n = 10)	1 (16.66%)	9 (64.28%)
Inappropriate empirical treatment	3 (50%)	3 (21.42%)
Targeted therapy		
Cefiderocol monotherapy	2 (16.66%)	6 (42.85%)
Cefiderocol-based regimen ^a^	1 (16.66%)	4 (28.57%)
Colistin monotherapy or based regimen ^b^	3 (50%)	3 (21.42%)
Polymicrobial infection	3 (50%)	7 (50%)
Recurrent CRAB infection	4 (66.6%)	1 (7.14%)
Invasive mycosis	2 (33.33%)	2 (14.28%)
AKI (infection related)	1 (16.66%)	2 (14.28%)
Early mortality (≤7 days)	0	
ICU infection related mortality	2 (33.33%)	
ICU mortality due to severity of burn injuries or decision to withhold or withdraw life-sustaining treatment because underlying conditions.	6 (100%)	

No differences have been made due to the small sample size. VAP: ventilator-associated pneumonia; BSI: bloodstream infection; AKI: acute kidney injury; ^a^: colistin or/and sulbactam; ^b^: sulbactam.

## Data Availability

The dataset analyzed during the current study is available from the corresponding author upon reasonable request.

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
