# Peer review of "Challenges Facing Two Outbreaks of Carbapenem-Resistant Acinetobacter baumannii: From Cefiderocol Susceptibility Testing to the Emergence of Cefiderocol-Resistant Mutants"

_antibiotics, 2024, doi:10.3390/antibiotics13080784_

Round 1

Reviewer 1 Report

Comments and Suggestions for Authors

The authors have conducted a study describing two nosocomial outbreaks of Carbapenem-resistant A. baumannii complex (CRAB).

Antimicrobial resistance is a silent pandemic of global scope. Cediferocol is a new antimicrobial whose resistance is modulated by a complex mutational resistome, which potentially confers cross-resistance to new combinations of beta-lactam beta-lactamases, as well as an extensive list of mutated iron uptake genes.

Observations:

The work is current and of great relevance for the treatment of infectious diseases. Some minor modifications are suggested (see file):

-It would be interesting to include in the discussion the issue of the inaccuracies of the techniques for determining susceptibility to cefiderocol. CLSI 2023 included a comment on the accuracy and reproducibility of cefiderocol susceptibility testing: "The accuracy and reproducibility of cefiderocol susceptibility testing by disk diffusion and broth microdilution are strongly affected by iron concentration and inoculum preparation, and may vary depending on the brand of paper disks and culture medium. Both false resistance and false susceptibility errors can occur. It is recommended that these false inaccuracies be discussed with the medical team."

-Tables should be read independently of the main text. Therefore, it is recommended to include information about the population, location, and period of the study

-Numbers should be written in letters at the beginning of a sentence

-Include italics in bacterial names

Author Response

Dear Colleague:

First of all, I would like to thank you for your positive assessment of the manuscript.

We have followed your suggestions: We have included in the discussion the issue of inaccuracies in the techniques for determining susceptibility to cefiderocol. We have added the CLSI and EUCAST alerts, and some recent works that we hope you will find interesting for the improvement of this manuscript.

We have also revised the text of the tables, the English spelling and included italics in the names of the microorganisms.

I hope that all these changes will be to your liking, and that we will finally have your approval for the publication of our manuscript in your journal.

Sincerely yours.

Dra. Montserrat Rodríguez Aguirregabiria

Critical Care Department. Hospital Universitario La Paz. Madrid. Spain.

Reviewer 2 Report

Comments and Suggestions for Authors

The paper focuses on a topic of interest, but several aspects need to be clarified or improved.

1. The title is not appropriate as the mechanisms of resistance to cefiderocol have not been investigated.

2. ‘Infection control interventions implemented during CRAB outbreaks in a single room ICU’ is not essential and may be removed by methods

3. the reference method for testing cefiderocol was not used. The discussion of the results obtained with disc diffusion and ComAsp must be included in the work by including recent published data. Is it possible to suggest an algorithm that includes both methods? Please discuss these points.

4. Synergy testing is described in Methods. Why? What did you test? Specify

5. In discussion, you introduce the role of combination of beta-lactamases inhibitors to enhance cefiderocol activity. Justify this statement by including published evidence (role of beta-lactamases on cefiderocol and synergy with beta-lactamase inhibitors) (see doi: 10.3390/antibiotics11121681).

Comments on the Quality of English Language

Minor editing of English language required.

Author Response

Dear Colleague:

First of all, I would like to thank you for your assessment of the manuscript and the suggestions you made with the purpose of improving it..

We have followed your suggestions:

  1. The title is not appropriate as the mechanisms of resistance to cefiderocol have not been investigated. Although we performed a phylogenetic analysis of the outbreak, the other problem we encountered was the issue of inaccuracies in the techniques for determining susceptibility to cefiderocol. We hope this one will be more appropriate for you.

“Challenges facing two outbreaks of carbapenem-resistant Acinetobacter baumannii: From cefiderocol susceptibility testing to the emergence of cefiderocol-resistant mutants.”

  1. ‘Infection control interventions implemented during CRAB outbreaks in a single room

ICU’ is not essential and may be removed by methods. Done.

  1. The reference method for testing cefiderocol was not used. The discussion of the results obtained with disc diffusion and ComAsp must be included in the work by including recent published data. Is it possible to suggest an algorithm that includes both methods?

BMD ComASP® was not available in our institution during first outbreak and the strains were tested afterwards. We have added the CLSI and EUCAST alerts, and some recent works focus on comparing the performance of different methods for testing in vitro activity of cefiderocol in CRAB isolates. We have rewritten this part of the discussion.

Kolesnik-Goldmann N, Seth-Smith HMB, Haldimann K, Imkamp F, Roloff T, Zbinden R, Hobbie SN, Egli A,  Mancini S. Com parison of Disk Diffusion, E-Test, and Broth Microdilution Methods for Testing In Vitro Activity of Cefiderocol in Acinetobacter baumannii. Antibiotics (Basel). 2023 Jul 20;12(7):1212.

Bianco G, Boattini M, Comini S, Banche G, Cavallo R, Costa C. Disc Diffusion and ComASP® Cefiderocol Microdilution Panel to Overcome the Challenge of Cefiderocol Susceptibility Testing in Clinical Laboratory Routine. Antibiotics (Basel). 2023 Mar 17;12(3):604.

Matuschek E, Longshaw C, Takemura M, Yamano Y, Kahlmeter G. Cefiderocol: EUCAST criteria for disc diffusion and broth microdilution for antimicrobial susceptibility testing. J Antimicrob Chemother. 2022 May 29;77(6):1662-1669.

Dortet L, Niccolai C, Pfennigwerth N, Frisch S, Gonzalez C, Antonelli A, Giani T, Hoenings R, Gatermann S, Rossolini GM, Naas T. Performance evaluation of the UMIC® Cefiderocol to determine MIC in Gram-negative bacteria. J Antimicrob Chemother. 2023, 78(7):1672-1676.

In terms of suggesting a strategy, in our experience disc diffusion could be useful for screening, and for resistant or uninterpretable isolates a BMD test should be performed to confirm the results.

  1. Synergy testing is described in Methods. Why? What did you test? Specify
  2. In discussion, you introduce the role of combination of beta-lactamases inhibitors to enhance cefiderocol activity. Justify this statement by including published evidence (role of beta-lactamases on cefiderocol and synergy with beta-lactamase inhibitors).

Since they are related we answer suggestions 4 and 5 together.

When we identified Acinetobacter baumannii strains resistant to cefiderocol, we reviewed the literature and found an interesting paper by Tamma PD, suggesting the combination with β-lactamase inhibitors, especially sulbactam in order to reduce the risk of in vivo emergence of cefiderocol-resistant strains or restore susceptibility in cefiderocol-resistant A. baumannii strains. Although the evidence is limited, we tested this synergy in one of the strains, specifically in a patient with a bacteremia. That strain was only susceptible to colistin, so when we found the synergy we treated the patient with cefiderocol, sulbactam and colistin. You can see it in the supplementary material. We have rewritten this part of the discussion and also added literature dealing precisely with this point.

Gill CM, Santini D, Takemura M, Longshaw C, Yamano Y, Echols R, Nicolau DP. In vivo efficacy & resistance prevention of cefiderocol in combination with ceftazidime/avibactam, ampicillin/sulbactam or meropenem using human-simulated regimens versus Acinetobacter baumannii. J Antimicrob Chemother. 2023 Apr 3;78(4):983-990.

Liu X, Lei T, Yang Y, Zhang L, Liu H, Leptihn S, Yu Y, Hua X. Structural Basis of PER-1-Mediated Cefiderocol Resistance and Synergistic Inhibition of PER-1 by Cefiderocol in Combination with Avibactam or Durlobactam in Acinetobacter baumannii. Antimicrob Agents Chemother. 2022 Dec 20;66(12):e0082822.

Mezcord V, Wong O, Pasteran F, Corso A, Tolmasky ME, Bonomo RA, Ramirez MS. Role of β-lactamase inhibitors on cefiderocol activity against carbapenem-resistant Acinetobacter species. Int J Antimicrob Agents. 2023 Jan;61(1):106700

Wong O, Mezcord V, Lopez C, Traglia GM, Pasteran F, Tuttobene MR, Corso A, Tolmasky ME, Bonomo RA, Ramirez MS. Hetero-antagonism of avibactam and sulbactam with cefiderocol in carbapenem-resistant Acinetobacter spp. bioRxiv [Preprint]. 2024 Mar 4:2024.03.04.583376.

I hope that all these changes will be to your liking, and that we will finally have your approval for the publication of our manuscript in your journal.

Sincerely yours.

Dra. Montserrat Rodríguez Aguirregabiria

Critical Care Department. Hospital Universitario La Paz. Madrid. Spain.

Round 2

Reviewer 2 Report

Comments and Suggestions for Authors

The paper has been improved and now it is suitable for publication.